# Anti-TCP1 Antibody Is a Potential Biomarker for Diagnosing Systemic Lupus Erythematosus

**DOI:** 10.3390/ijms25168612

**Published:** 2024-08-07

**Authors:** Sang-Won Lee, Wook-Young Baek, So-Won Park, Jee-Min Chung, Ji-Hyun Park, Ho Chul Kang, Ju-Yang Jung, Chang-Hee Suh

**Affiliations:** 1Department of Rheumatology, Ajou University School of Medicine, Suwon 16499, Republic of Korea; znfdla@naver.com (S.-W.L.); arikato83@naver.com (W.-Y.B.); sowon9413@naver.com (S.-W.P.); serinne20@hanmail.net (J.-Y.J.); 2Department of Molecular Physiology, Ajou University School of Medicine, Suwon 16499, Republic of Korea; cjm316555@gmail.com (J.-M.C.); hckang@ajou.ac.kr (H.C.K.); 3Office of Biostatistics, Medical Research Collaborating Center, Ajou Research Institute for Innovative Medicine, Ajou University Medical Center, Suwon 16499, Republic of Korea; jhn1105@gmail.com; 4Department of Mathematics, Ajou University, Suwon 16499, Republic of Korea; 5Department of Molecular Science and Technology, Ajou University, Suwon 16499, Republic of Korea

**Keywords:** systemic lupus erythematosus, autoantibody, protein chip, chaperonin containing t-complex polypeptide 1, biomarker

## Abstract

Systemic lupus erythematosus (SLE) is a chronic inflammatory disease caused by autoantibodies. Serum samples from patients with SLE (*n* = 10) were compared with those from normal controls (NCs, *n* = 5) using 21K protein chip analysis to identify a biomarker for SLE, revealing 63 SLE-specific autoantibodies. The anti-chaperonin-containing t-complex polypeptide-1 (TCP1) antibody exhibited higher expression in patients with SLE than in NCs. To validate the specificity of the anti-TCP1 antibody in SLE, dot blot analysis was conducted using sera from patients with SLE (*n* = 100), rheumatoid arthritis (RA; *n* = 25), Behçet’s disease (BD; *n* = 28), and systemic sclerosis (SSc; *n* = 30) and NCs (*n* = 50). The results confirmed the detection of anti-TCP1 antibodies in 79 of 100 patients with SLE, with substantially elevated expression compared to both NCs and patients with other autoimmune diseases. We performed an enzyme-linked immunosorbent assay to determine the relative amounts of anti-TCP1 antibodies; markedly elevated anti-TCP1 antibody levels were detected in the sera of patients with SLE (50.1 ± 17.3 arbitrary unit (AU), *n* = 251) compared to those in NCs (33.9 ± 9.3 AU), RA (35 ± 8.7 AU), BD (37.5 ± 11.6 AU), and SSc (43 ± 11.9 AU). These data suggest that the anti-TCP1 antibody is a potential diagnostic biomarker for SLE.

## 1. Introduction

Autoimmune diseases occur when an abnormal inflammatory immune response occurs in the body, affecting tissue, organs, or cells [1]. Although it is understood that autoimmune diseases are influenced by genetic, hormonal, infectious, and environmental factors, the exact mechanism remains unclear [2]. A common characteristic of autoimmune diseases is the excessive secretion of autoantibodies, which can lead to immune disorders by forming autoantibodies and immune complexes [3]. These complexes accumulate in tissues, causing inflammation, which is critical in pathogenesis [4]. Systemic lupus erythematosus (SLE), rheumatoid arthritis (RA), Behçet’s disease (BD), and systemic sclerosis (SSc) are among the most prominent autoimmune diseases. SLE was diagnosed by the 2019 American College of Rheumatology/European League Against Rheumatism classification criteria. These criteria were developed to improve the diagnostic sensitivity and specificity. A positive antinuclear antibody test result is a mandatory entry criterion, and SLE is classified by accumulating at least 10 points in the clinical and immunological domains. The clinical domains included fever, leukopenia, thrombocytopenia, neuropsychiatric symptoms, and renal involvement. Immunological domains include autoantibodies, including highly specific antibodies, such as anti-dsDNA and anti-Sm antibodies. Autoantibody testing is essential to ensure a comprehensive and accurate diagnosis of SLE [5,6,7,8,9].

Biological biomarkers are cellular, biological, or biochemical variations that can be measured in biological materials such as human cells, tissues, or body fluids [10]. Various biomarkers have been identified to date, including proteins such as antigens or antibodies; nucleic acids such as DNA, microRNAs, and non-coding RNAs; and protein posttranslational modifications. These biomarkers are routinely used for the clinical diagnosis of various diseases [11].

Microarray technology involves miniaturizing thousands of DNA molecules, proteins, cells, and other components onto slides. Protein microarrays have emerged as highly efficient tools for addressing the limitations of DNA microarrays. They offer a direct platform for the analysis of protein functions. Protein microarrays are created by immobilizing all proteins encoded by an organism, making them valuable tools for investigating protein functions such as protein–protein interactions, biochemical activities, enzyme–substrate relationships, and immune responses [12,13].

SLE is a complex autoimmune disease characterized by the production of autoantibodies against various cellular components [14]. In this context, identifying specific autoantibodies associated with SLE can provide valuable insights into its pathogenesis and potential biomarkers for diagnosis.

The chaperonin-containing t-complex polypeptide 1(TCP1) plays pivotal roles in intricate gene expression, protein folding, and assembly processes. In contrast to other chaperones, TCP1 is localized in the cytoplasm of eukaryotic cells. Its structure embodies a hetero-oligomeric configuration comprising no fewer than eight distinct subunit species [15]. These subunits share crucial motifs essential for ATPase function, suggesting a duality of specific and common functions within each unit [16]. As a unified entity, TCP1 orchestrates a sophisticated mechanism for protein folding and assembly within nucleated cells, and chaperone activity is indispensable for the proper folding and assembly of diverse synthetic polypeptides [17,18]. There is a lack of research on the anti-TCP1 antibody in SLE, and its role in SLE pathogenesis remains unclear.

This study aimed to identify the biomarkers unique to SLE. The sera of patients with SLE and healthy individuals were analyzed and compared using a 21K human proteome microarray. Using a microarray approach, we initially identified 63 SLE-specific autoantibody candidates. Among these, we prioritized genes that ranked highly in our analysis. Genes with substantial sizes were excluded from further investigation, which resulted in the anti-TCP1 antibody emerging as a notable candidate. Within the selected autoantibodies, the anti-TCP1 antibody showed significantly elevated expression in SLE. Therefore, we aimed to evaluate the potential of the anti-TCP1 antibody as a promising biomarker for SLE.

To determine whether an anti-TCP1 antibody could serve as a potential biomarker for SLE, this study aims to elucidate the potential role of anti-TCP1 antibody as a biomarker in SLE, contributing to a deeper understanding of the disease mechanisms and paving the way for future research in autoimmune diagnostics and therapeutics. We generated Glutathione S-transferases (GST)-TCP1 protein. We then analyzed serum samples from patients with SLE, NCs, and various autoimmune diseases such as RA, BD, and SSc through dot blot analysis and enzyme-linked immunosorbent assay (ELISA) (Figure 1).

## 2. Results

### 2.1. SLE Autoantibody Selection Using 21K Protein Chip 

To identify a specific biomarker for SLE, serum samples from 10 patients with SLE and 5 NCs were reacted on a 21K protein chip and probed with anti-human IgG antibodies to analyze the immune response. To identify autoantibodies specific to SLE, we excluded antigens that exhibited positive signals from NCs. Through a comparative analysis of protein chips using Microscan (Molecular Devices, San Jose, CA, USA), 63 SLE-specific autoantibodies were identified (Figure 2). Among these, anti-60S acidic ribosomal protein P0, P1, and P2 (RPLP0, RPLP1, and RPLP2) antibodies, which are autoantibodies against SLE (16), were confirmed. Anti-RPLP antibodies were detected in samples from four of the ten patients with SLE. Additionally, the anti-TCP1 antibody was detected in eight samples from patients with SLE.

### 2.2. GST Fusion Protein Construction 

We amplified the *TCP1* gene (Harvard Plasmid, Cambridge, MA, USA) by polymerase chain reaction using designed primers to produce a GST fusion protein. The *TCP1* gene length was 1671 bp and was confirmed by agarose gel electrophoresis (Figure 3A). Using the Bac-to-Bac system, we produced a Bacmid containing TCP1. We then transfected Sf9 cells with Bacmid to produce the GST-TCP1 protein (85 kDa). Production of GST-TCP1 was confirmed by Western blot analysis using an anti-GST antibody. We infected Sf9 cells with baculovirus to express the GST-TCP1 protein and then purified the protein. GST-RPLP0 (60 kDa), GST-RPLP1 (38 kDa), and GST-RPLP2 (38 kDa) proteins were produced using the same procedure as for GST-TCP1. The proteins were confirmed by Coomassie staining after electrophoresis (Figure 3B,C).

### 2.3. The Expression of Anti-TCP1 Antibody in Patients with SLE in the Dot Blot

We performed a dot blot assay using sera from 100 patients with SLE. Anti-RPLP0, anti-RPLP1, and anti-RPLP2 were used as controls. We coated nitrocellulose membranes with GST, GST-RPLP0, GST-RPLP1, and GST-RPLP2 proteins and exposed them to sera from patients with SLE. We observed the co-expression of anti-RPLP0, RPLP1, and RPLP2 antibodies. It was detected in 51 of 100 patient samples (Figure 4A). We coated GST and GST-TCP1 proteins and performed dot blotting in patients with SLE. It was confirmed through dot blot results that the anti-TCP1 antibody was expressed in 79 out of 100 patients (Figure 4B).

### 2.4. Expression of Anti-TCP1 Antibody in Patients with Other Autoimmune Diseases in the Dot Blot

We confirmed that anti-TCP1 antibodies were highly expressed in SLE through dot blot analysis. We performed dot blot assays using sera from NCs and patients with RA, BD, or SSc to validate this finding. Dot blot analysis revealed anti-TCP1 antibody expression in only one of the 50 NCs. Among the 25 patients with RA, there was no expression of anti-TCP1 antibodies; however, 5 of 28 patients with BD and 5 of 30 patients with SSc showed anti-TCP1 antibody expression. Additionally, expression was observed in all serum samples from the five patients with SLE and used as positive controls (Figure 5A–E). When comparing SLE with NCs, the sensitivity of the anti-TCP1 antibody was 79% (95% CI: 69.7–86.5%) and the specificity was 98% (95% CI: 89.4–99.9%). When comparing SLE with NCs and other autoimmune disease (RA, BD and SSc), the sensitivity was the same, and the specificity was 91.7% (95% CI: 85.7–95.8%).

### 2.5. Anti-TCP1 Antibody Expression in the ELISA

We conducted an ELISA to quantitatively confirm the specific expression of anti-TCP1 antibodies in SLE. We coated GST-TCP1 and utilized serum samples from patients with SLE (*n* = 251), RA (*n* = 25), BD (*n* = 28), SSc (*n* = 30), and NCs (*n* = 50). In the ELISA results, the average levels of anti-TCP1 antibody were significantly higher in patients with SLE (50.1 ± 17.3 arbitrary unit (AU)) compared to those of RA (35 ± 8.7 AU), BD (37.5 ± 11.6 AU), SSc (43 ± 11.9 AU), and NCs (33.9 ± 9.3 AU) (Figure 6).

## 3. Discussion

Protein chip microarrays are an excellent experimental technique for identifying interactions between proteins, enzyme–substrate interactions, antibody–antigen reactions, and finding biomarkers [13]. Currently, SLE is comprehensively diagnosed based on several clinical and immunological criteria. Autoantibodies contribute significantly to organ damage and serve as a pivotal focus in disease management [19,20]. This study aimed to identify SLE-specific autoantibodies and provide a more precise diagnosis. To identify specific autoantibodies in the sera of patients with SLE, we used a protein chip microarray containing 21,000 purified human proteins and found 63 SLE-specific autoantibodies compared to those in NCs. According to the protein chip results, anti-RPLP0, RPLP1, and RPLP2 antibodies, already known as autoantibodies in SLE and showing positive reactions in 40% of cases, were used as confirmation [21,22]. Among the 63 identified autoantibodies, we focused on selecting genes with high expression on the microarray. We considered the expression efficiency when producing GST fusion protein, recognizing that genes with larger sizes might not express well. Therefore, we excluded these larger genes from further investigation. This careful selection process led us to identify the anti-TCP1 antibody as a notable candidate due to its 80% positive reaction rate in SLE patients on microarray. To confirm the presence of autoantibodies in the sera of SLE patients, we purified GST fusion proteins for RPLP0, RPLP1, RPLP2, and TCP1.

RPLP, a 60S acidic ribosomal protein, possesses several motifs involved in recruiting ribosome-inactivating proteins to the C-terminal region of the ribosomal stalk; however, its biological functions remain unclear. In eukaryotes, RPLP0, RPLP1, and RPLP2 are pivotal components of the ribosomal stalk that are crucial for regulating ribosome activity and exhibiting tissue-specific expression patterns. Implicated in diseases like autoimmune disorders and cancer, they potentially contribute to tumorigenesis and metastasis, particularly RPLP1, which is associated with cell cycle regulation and malignant transformation [23]. In another study of anti-RPLP0, RPLP1, and RPLP2 antibodies, known as NPSLE biomarkers, the sensitivity was 26% (95% CI: 15–42%), and the specificity was 80% (95% CI: 74–85%) [24].

The dot blot analysis revealed the simultaneous expression of antibodies against GST-RPLP0, RPLP1, and RPLP2 proteins, detected in 51 out of 100 serum samples from patients with SLE. Anti-TCP1 antibodies were detected in 79 of 100 serum samples from patients with SLE, showing a higher expression rate than anti-RPLP antibodies. Both the microarray and dot blot assay results showed that patients with SLE simultaneously expressed all three antibodies (RPLP0, RPLP1, and RPLP2) in the serum. In another study, RPLP0, RPLP1, and RPLP2 were assembled together on the 28 S ribosomal RNA. When the N-terminal 10 amino acids of RPLP1 or RPLP2 were deleted, their binding to RPLP0 was disrupted [25]. Microarray and dot blot analyses have revealed that autoantibodies against all three RPLP proteins are observed simultaneously in SLE sera. This phenomenon can be attributed to epitope spreading, where an initial immune response against a single protein component leads to the generation of autoantibodies against multiple proteins within the same complex [26]. Analysis of the results of the microarray and dot blot experiments revealed that the anti-TCP1 antibody showed an expression rate of 80% in the microarray and 79% in the dot blot, whereas the anti-RPLP antibodies showed an expression rate of 40% in the microarray and 51% in the dot blot. This indicates that the anti-TCP1 antibody is more prevalent in the serum of SLE patients compared to the three anti-RPLP antibodies, suggesting that anti-TCP1 could serve as a potential biomarker for SLE.

Next, we hypothesized that the anti-TCP1 antibody could be a better biomarker than the anti-RPLP0, RPLP1, and RPLP2 antibodies, which were already known lupus autoantibodies. To confirm whether the anti-TCP1 antibody was expressed explicitly in SLE, dot blot analysis was performed on control groups with other autoimmune diseases and NCs. The anti-TCP1 antibody was detected in only 1 of 50 of the NCs. None of the 25 patients with RA expressed anti-TCP1 antibodies. However, 5 of 28 patients with BD and 5 of 30 patients with SSc expressed anti-TCP1 antibodies. The reason anti-TCP1 antibodies were detected in some BD and SSc patients appears to be that autoimmune diseases, including SLE, produce a variety of autoantibodies. Although the number of samples was small, the anti-TCP1 antibody appears to have been detected in dot blots from patients with BD and SSc. When comparing the sensitivity and specificity of the anti-TCP1 antibody between SLE and NCs, the sensitivity was 79% and the specificity was 98%. When comparing SLE with NCs and other autoimmune diseases (RA, BD and SSc), the sensitivity was the same, and the specificity was 91.7%. This suggests that the anti-TCP1 antibody could be a reliable biomarker for distinguishing SLE from other autoimmune diseases and NCs.

We conducted ELISA to compare the relative levels of anti-TCP1 antibodies between patients with SLE and control groups. ELISA results were measured in the AU, with the sample with the highest optical density (OD) value among the SLE sera used as the reference point. Anti-TCP1 antibody levels were significantly higher in patients with SLE than in those with RA, BD, SSc, or in NCs. Although dot blot results showed some anti-TCP1 antibody expression in patients with BD and SSc, ELISA revealed higher anti-TCP1 antibody levels in the sera of patients with SLE. It has been suggested that anti-TCP1 antibody levels are significantly elevated in SLE and may be helpful in the diagnosis of SLE.

In this study, the expression level of the anti-TCP1 antibody was confirmed in each group. Because the standard of anti-TCP1 antibody is not available, the results were expressed in arbitrary units. Further research is needed to identify more specific and sensitive antigenic epitopes to enhance the sensitivity of the ELISA method. Additionally, SLE patients were not classified according to disease activity, which is a significant limitation. Consequently, follow-up studies are needed to address this gap. Additionally, these studies did not include a detailed correlation analysis between ELISA measurements and the Systemic Lupus Erythematosus Disease Activity Index (SLEDAI). Therefore, further research should also explore the potential mechanisms by which the anti-TCP1 antibody contributes to SLE pathogenesis, including its role in immune regulation and autoantibody production.

In summary, this study identified the anti-TCP1 antibody as a potential biomarker for SLE, showing higher expression rates than the known anti-RPLP0, RPLP1, and RPLP2 antibodies. The anti-TCP1 antibody exhibited specificity for SLE with significantly elevated levels in patient sera, as confirmed by dot blot and ELISA analyses. This study suggests that the anti-TCP1 antibody is a promising diagnostic tool for SLE. However, further research is needed to validate its clinical utility and potential for assessing disease activity.

## 4. Materials and Methods

### 4.1. Patients and Samples

After obtaining informed consent, blood and serum samples were acquired from 251 Korean patients with SLE who met the revised diagnostic criteria outlined in the 2019 ACR/EULAR classification [9]. This study also included a control group of 50 NCs comprising 25 patients with RA who met the 2010 ACR/EULAR classification criteria [27], 28 patients with BD who met the 1990 International Study Group criteria [28], and 30 patients with SSc who met the 2013 ACR/EULAR classification criteria [29]. For all groups, patients showing symptoms of other autoimmune diseases or conditions were excluded (Appendix A). Ethical clearance for the study was granted by the IRB (AJOUIRB-OBS-2015-423) at Ajou University Hospital, and all participants provided informed consent and were thoroughly explained the objectives of the study.

### 4.2. Human Protein Microarray

Human protein microarray assays using the HuProt human proteome microarray v3.0 (CDI Laboratories Inc., Mayaguez, PR, USA), which contains a comprehensive collection of over 22,000 full-length human proteins, were performed as previously described by Chung et al. [30]. In total, 21K microarrays were incubated with a microarray buffer (137 mM NaCl, 2.7 mM KCl, 4.3 mM Na_2_HPO_4_, 1.8 mM KH_2_PO_4_, pH 7.4, and 0.05% Triton X-100) for 5 min at 20–24 °C. To reduce background signals, arrays were blocked with 5% IgG free BSA (Merck KGaA, Darmstadt, Germany) in microarray buffer for an hour at 20–24 °C, followed by incubation with 200 ug of serum in a reaction buffer (50 mM Tris-Cl, pH 7.5; 2 mM DTT; 2.5 mM MgCl_2_) overnight at 4 °C. Subsequently, arrays were washed with microarray buffer and incubated with Alexa Fluor 647-conjugated goat anti-human IgG (1:5000 dilution) (Invitrogen, Carlsbad, CA, USA). After washing with microarray buffer, arrays were dried and immediately scanned using an Axon GenePix 4000B microarray scanner (Molecular Devices, San Jose, CA, USA). All proteins on the chip were probed using GST antibodies and Alexa Fluor 546-conjugated goat anti-rabbit secondary antibodies (1:5000 dilution) (Invitrogen, Carlsbad, CA, USA). The signal intensity for each spot was recorded as the ratio of foreground to background signal and normalized to the signal of GST using the Axon GenePix 4000B microarray scanner. Finally, the mean signal intensity of all proteins on the chip was calculated. To determine the cut-off value for highly expressed autoantibodies, we calculated the amount of proteins with signal intensities greater than the mean + 2 SD, which were considered highly expressed. This cut-off was chosen to identify proteins with significantly elevated expression levels, ensuring specificity for SLE autoantibodies. Proteins exceeding this threshold were flagged for further analysis. Using these established experimental conditions, SLE-specific autoantibodies were observed in the sera of patients with SLE (*n* = 10) compared to controls (NCs, *n* = 5).

### 4.3. Polymerase Chain Reaction (PCR)

Primer design was based on the *TCP1* gene sequence registered in NCBI, and the *TCP1* gene was amplified using the following TCP1 primers: 5′-ATGGAGGGGCCTTTGTCCGTGTT-3′ (forward primer) and 5′-TCACAAATCGTTAAGGGCTCCAGAGTG-3′ (reverse primer).

### 4.4. Cloning

The TA cloning process was conducted using a TA cloning kit (Invitrogen, Carlsbad, CA, USA), following the manufacturer’s guidelines. The gateway cloning process was conducted using Gateway™ LR Clonase™ II Enzyme mix (Thermo, Waltham, MA, USA), following the manufacturer’s guidelines. 

### 4.5. Plasmid DNA Extraction and Sequencing

DNA was extracted using the Plasmid Mini-Prep Kit (BIOFACT, Daejeon, Chungcheongnam-do, South Korea) according to the manufacturer’s protocol. The extracted DNA was stored at −70 °C. Orientation was confirmed using the MacroGen standard sequencing service (Macrogen, Seoul, South Korea). A commercially available M13F primer was used for the analysis.

### 4.6. Inserting Recombinant DNA Using the Bac-to-Bac System and Bacmid Extraction

The recombinant DNA was transformed into DH10BAC *E. coli* cells using the Bac-to-Bac system. Transformed *E. coli* cells were selected using a medium containing kanamycin, tetracycline, and gentamicin. For DNA extraction, the Phase-Prep BAC DNA Kit (Sigma-Aldrich Corporation, St. Louis, MO, USA) was used following the manufacturer’s protocol. The concentration of extracted DNA was assessed using a NanoDrop Lite spectrophotometer (Thermo, Waltham, MA, USA).

### 4.7. Transfection

Bacmid and Cellfectin II reagent kits (Invitrogen, Carlsbad, CA, USA) were mixed and transfected into SF9 cells, which were further incubated at 27 °C for 5 days. Transfected cells for protein purification were harvested on day 3. The cells were centrifuged, and the cell pellets were stored at −70 °C, whereas the supernatant was stored at 4 °C.

### 4.8. Western Blotting Assay

Sodium dodecyl sulfate–polyacrylamide gel electrophoresis was conducted, and the proteins were transferred onto a polyvinylidene fluoride membrane. Next, 5% skim milk (0.1% TBST) was added, and the mixture was allowed to react for 1 h at 20–24 °C. The primary antibody used was the anti-GST antibody (Abcam, Cambridge, MA, UK) at a dilution of 1:5000 in 1% skim milk (0.1% TBST). The secondary antibody used was an anti-human antibody (Abcam, Cambridge, MA, UK) diluted 1:10,000 in 1% skim milk (0.1% TBST), followed by detection using Western Bright ECL (Advansta, San Jose, CA, USA).

### 4.9. GST-Fused Protein Purification

The cells were lysed using NETN buffer. GST-TCP1 was selected with Glutathione Sepharose (GE Healthcare, Chicago, IL, USA) beads, and the purified protein was stored at −70 °C.

### 4.10. Dot-Blotting Assay

GST and GST fusion proteins were coated onto nitrocellulose membranes and incubated at 20–24 °C for 10 min. A blocking buffer containing 5% BSA (0.1% PBST) was added and incubated for 30 min at 20–24 °C. The membrane was washed with 0.1% PBST. Serum samples from the control group or patient groups were added and incubated overnight at 4 °C. The secondary antibody, a goat anti-human antibody (Abcam, Cambridge, MA, UK) conjugated with HRP, was diluted in 1% BSA buffer (0.1% PBST) at a ratio of 1:5000 and incubated at 20–24 °C for 1 h. The membrane was then exposed to SuperSignal West Pico Chemiluminescent Substrate (Thermo Scientific, Waltham, MA, USA) and developed in a dark room for an appropriate duration to assess the levels of autoantibodies in the serum.

### 4.11. ELISA

The GST fusion protein was coated onto each well of the plate at a concentration of 200 ng per well and incubated overnight at 4 °C. Blocking was performed using 5% BSA in PBS to prevent non-specific binding. Human serum samples were diluted 1:1000 in PBS. The secondary antibody, anti-human IgG (Abcam, Cambridge, UK), was diluted 1:10,000 in PBS. The reaction was developed using OptEIA TMB Substrate Reagent (BD Biosciences, Franklin Lakes, NJ, USA) in a light-blocked environment and allowed to proceed for 5 min at 20–24 °C. The reaction was stopped using 2 N sulfuric acid. Absorbance at 450 nm was determined using an iMark™ Microplate Absorbance Reader (Bio-rad, Hercules, CA, USA). The percentage of AU was calculated by selecting the SLE sample with the highest OD for the anti-TCP1 antibody.

### 4.12. Statistical Analysis

For continuous variables, the Kolmogorov–Smirnov test was used to verify normality before the analysis, and quantitative data with normal distributions were expressed as the mean ± standard deviation. Differences in variables between groups were evaluated using the Kruskal–Wallis test or Student’s *t*-test was used for continuous variables, and the Chi-square test for categorical variables. To evaluate the diagnostic ability of the anti-TCP1 antibody identified by dot blot, sensitivity and specificity with 95% confidence intervals were calculated. To compare differences in serum anti-TCP1 antibody expression measured by ELISA, average AU values were evaluated and the Kruskal–Wallis test was used for data with nonparametric distribution. Statistical significance was defined as a *p* value of <0.05 for all analyses. Statistical analyses were performed using R version 4.3.3 (R Project for Statistical Computing).

## Figures and Tables

**Figure 1 ijms-25-08612-f001:**
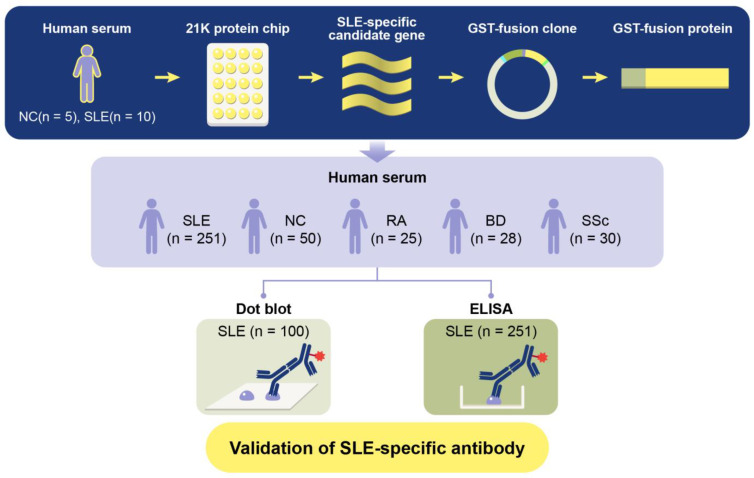
Schematic diagram of GST fusion protein production process and verification experiment for systemic lupus erythematosus (SLE)-specific autoantibody candidates.

**Figure 2 ijms-25-08612-f002:**
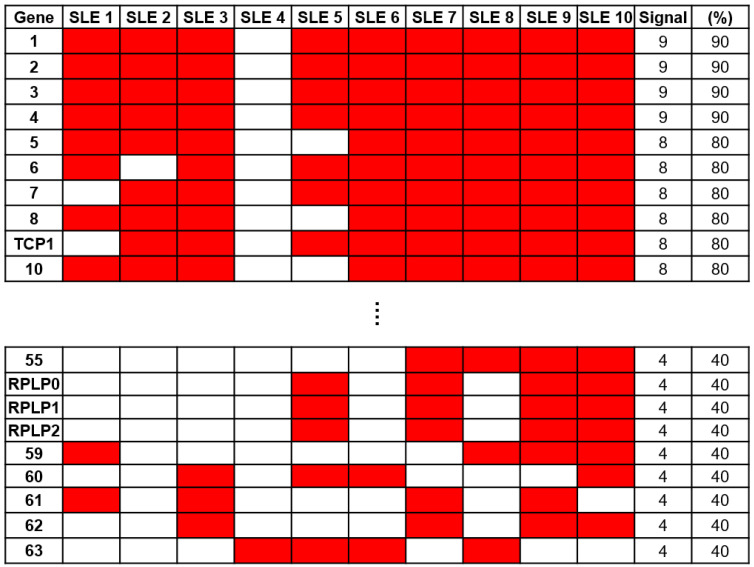
Screening of autoantibody samples using 21K protein. Sixty-three systemic lupus erythematosus (SLE)-specific autoantibodies were selected through comparative analysis using sera from patients with SLE (*n* = 10) and normal controls (NCs) (*n* = 5) on a 21K protein chip.

**Figure 3 ijms-25-08612-f003:**
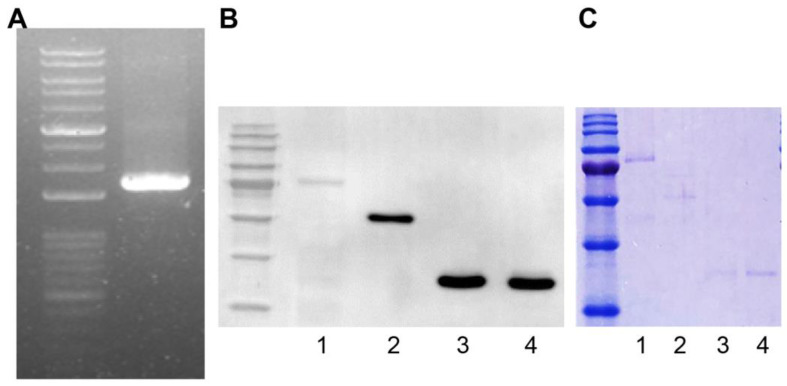
Production of the GST fusion protein for identifying systemic lupus erythematosus (SLE)-specific autoantibodies. (**A**) TCP1 (1671 bp) gene amplification by polymerase chain reaction. (**B**) Confirmation of GST-TCP1 expression in SF9 cells by Western blot. First antibody: rabbit GST antibody (1:5000); second antibody: goat anti-rabbit HRP-conjugated antibody (1:10,000). 1. GST-TCP1, 2. GST-RPLP0 (60 kDa), 3. GST-RPLP1 (38 kDa), 4. GST-RPLP2 (38 kDa). (**C**) Identification of produced GST fusion protein on 12% sodium dodecyl sulfate–polyacrylamide gel using Coomassie staining. 1. GST-TCP1, 2. GST-RPLP0 (60 kDa), 3. GST-RPLP1 (38 kDa), 4. GST-RPLP2 (38 kDa).

**Figure 4 ijms-25-08612-f004:**
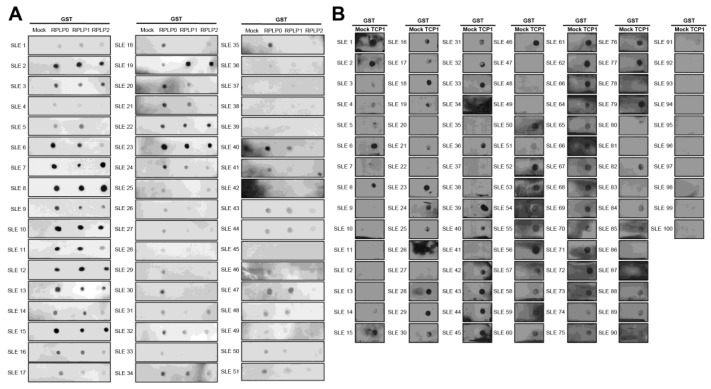
Dot blot analysis of anti-RPLPs and anti-TCP1 antibodies using sera from patients with systemic lupus erythematosus (SLE). Coating GST, GST-RPLPs and GST-TCP1 proteins on nitrocellulose membrane: (**A**) analysis of anti-RPLP antibodies. (**B**) analysis of anti-TCP1 antibody in SLE sera (*n* = 100) using dot blot. Second antibody: goat anti-human HRP conjugated antibody (1:10,000).

**Figure 5 ijms-25-08612-f005:**
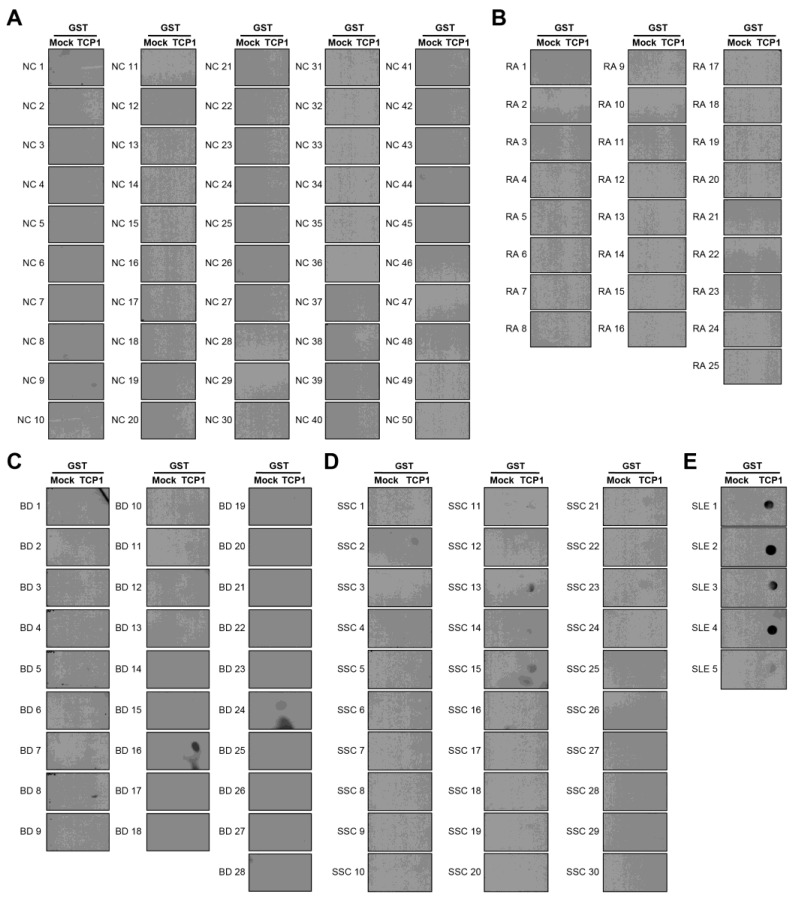
Dot blot analysis of anti-TCP1 antibodies using sera from normal controls (NCs) and patients with other autoimmune diseases. Coating GST and GST-TCP1 proteins on nitrocellulose membrane: (**A**) analysis of anti-TCP1 antibody in NCs (*n* = 50), (**B**) rheumatoid arthritis (RA) (*n* = 25), (**C**) Behçet’s disease (BD) (*n* = 28), (**D**) systemic sclerosis (SSc) (*n* = 30), (**E**) SLE (*n* = 5) sera using dot blot. Second antibody: goat anti-human HRP conjugated antibody (1:10,000).

**Figure 6 ijms-25-08612-f006:**
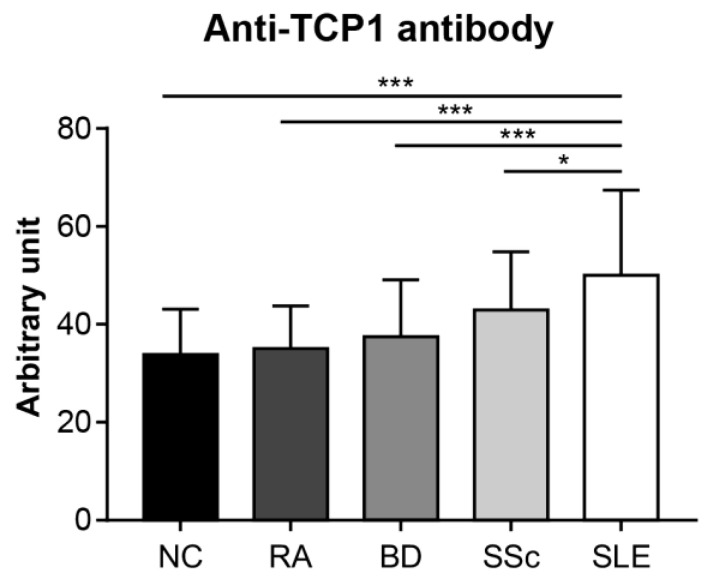
Measurement of the relative amount of serum anti-TCP1 antibody in patients with systemic lupus erythematosus (SLE) and control groups using enzyme-linked immunosorbent assay. Quantitative measurements of anti-TCP1 antibody in sera from normal controls (NCs) (*n* = 50), patients with rheumatoid arthritis (RA) (*n* = 25), Behçet’s disease (BD) (*n* = 28), systemic sclerosis (SSc) (*n* = 30), and SLE (*n* = 251) were performed in duplicate wells. Statistical analyses were performed using R software (version 4.3.3). The Kruskal–Wallis test was used for continuous variables with nonparametric distribution (* *p* < 0.05, *** *p* < 0.001).

## Data Availability

The data presented in this study are available on request from the corresponding author.

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
