# Peer review of "Anti-TCP1 Antibody Is a Potential Biomarker for Diagnosing Systemic Lupus Erythematosus"

_ijms, 2024, doi:10.3390/ijms25168612_

Round 1

Reviewer 1 Report

Comments and Suggestions for Authors

This article investigates the potential of anti-TCP1 (T-complex protein 1) antibody as a novel biomarker for diagnosing systemic lupus erythematosus (SLE). The authors used a protein microarray approach to screen for autoantibodies in SLE patient sera, identifying anti-TCP1 as a candidate. They then validated this finding using dot blot and ELISA techniques, comparing anti-TCP1 antibody levels in SLE patients to those with other autoimmune diseases and healthy controls. While the study presents an interesting potential new biomarker for SLE, there are significant issues with methodology, data presentation, and overall quality that need to be addressed before publication can be considered. Detailed comments for improvement:
1. The introduction lacks sufficient background on TCP1 and its potential relevance to SLE pathogenesis. More context is needed to justify why this protein was selected for study.
2. The methods section requires significant expansion and clarification:
a) Patient selection: Details on patient selection criteria and demographics are lacking. The authors should provide information on how SLE patients were diagnosed, inclusion/exclusion criteria, and basic demographic data (age, sex, disease duration, etc.) for all groups studied.
b) Protein chip methodology: The description of the 21K protein chip analysis is very brief and lacks important details. The authors should provide information on:
- How the chips were prepared and what proteins were included
- How positive signals were defined and quantified
- What software was used for analysis and how cutoffs for significance were determined
c) Statistical analysis: The statistical methods are not clearly explained. The authors state they used R software, but do not specify what tests were performed for which comparisons.
3. The discussion is limited and does not adequately interpret the results in context of existing literature. Potential mechanisms and clinical implications should be explored further. What are the limitations of this study and what further research is needed to validate anti-TCP1 as a diagnostic biomarker for SLE?

Major revisions focusing on improved methods description, statistical analysis, and interpretation of results in context of existing literature are strongly recommended. Additionally, the manuscript would benefit greatly from thorough editing for English language and grammar.

Comments on the Quality of English Language

The manuscript would benefit from some additional editing to enhance the clarity and fluency of the English language. While the overall meaning is generally clear, there are areas where the phrasing could be more precise or idiomatic.  For example:
In the sentence "Sixty-three sys- temic lupus erythematosus (SLE)-specific autoantibodies selected through comparative analysis...", a verb is missing. This could be improved to: "Sixty-three systemic lupus erythematosus (SLE)-specific autoantibodies were selected through comparative analysis..."
The phrase "We confirmed that anti-RPLP0, RPLP1, and RPLP2 antibodies were coexpressed" could be rephrased for clarity as: "We observed the co-expression of anti-RPLP0, RPLP1, and RPLP2 antibodies."

Reviewer 2 Report

Comments and Suggestions for Authors

"This study aimed to identify the biomarkers unique to SLE." The authors assessed, by laboratory methods I am not competent to judge the correctness, an antiTCP1 antibody selected from 21000 candidates, comparing 10 patients with SLE with 5 normal controls. This anti-TCP1 was detected in the serum of 8 of the 10 patients with SLE and in none of the 5 healthy controls. Therefore, the sensitivity of this antibody was 80%, with a calculated 95%CI of 73-86%, while the specificity could be 100%, but taking into account the small sample size of only 5 NC, the 95%CI of Specificity is calculated to be from 55 to 100%.

Moreover, when tested in a sample of other autoimmune diseases, anti-TCP1 was present in all the diseases, with a lower mean titer than in patients with SLE ("SLE 196 (50.1 ± 17.3 arbitrary unit (AU)) than those of RA (35 ± 8.7 AU), BD (37.5 ± 11.6 AU), SSc 197 (43 ± 11.9 AU), and NCs (33.9 ± 9.3 AU)"),.

Therefore, without being able to judge the laboratory methodology, anti-TCP1 could be a valuable tool for the diagnosis of SLE. However, from this study, the sensitivity=80%, 95%CI [73-86%], while the specificity has to be determined - the titers are statistically different, but the range of values from NCs and different autoimmune diseases intersects those of SLE patients, therefore the diagnostic value has to be determined.

These observations should be contained in the Discussion section.

Round 2

Reviewer 2 Report

Comments and Suggestions for Authors

I am satisfied with the improvements.